# Use of Intraoral Scanners for Full Dental Arches: Could Different Strategies or Overlapping Software Affect Accuracy?

**DOI:** 10.3390/ijerph18199946

**Published:** 2021-09-22

**Authors:** Luigi Vito Stefanelli, Alessio Franchina, Andrea Pranno, Gerardo Pellegrino, Agnese Ferri, Nicola Pranno, Stefano Di Carlo, Francesca De Angelis

**Affiliations:** 1Department of Oral and Maxillofacial Sciences, Sapienza University of Rome, Via Caserta 6, 00161 Rome, Italy; gigistef@libero.it (L.V.S.); stefano.dicarlo@uniroma1.it (S.D.C.); francesca.deangelis@uniroma1.it (F.D.A.); 2Private Practice, Via Legione Gallieno 44, 36100 Vicenza, Italy; alessiofranchina@icloud.com; 3Department of Civil Engineering, University of Calabria, 87036 Rende, Italy; andrea.pranno@unical.it; 4Unit of Oral and Maxillofacial Surgery, Department of Biomedical and Neuromotor Sciences, University of Bologna, 40121 Bologna, Italy; gerardo.pellegrino2@unibo.it (G.P.); agnese.ferri3@unibo.it (A.F.)

**Keywords:** intraoral scanner, digital impression, digital dentistry, trueness, precision, accuracy

## Abstract

Objectives: The use of digital devices is strongly influencing the dental rehabilitation workflow both for single-crown rehabilitation and for full-arch prosthetic treatments. Methods: In this study, trueness was analyzed by overlapping the scan dataset made with Medit I-500 (by using two different tips and two different scan strategies) with the scan dataset made with lab scanning, and the values of the (90°–10°)/2 method were reported. Precision was evaluated by using the same values of trueness coming from the intra-group overlapping (scan dataset made with an IOS overlapped and compared to each other). Moreover, two different software programs of overlapping were used to calculate accuracy values. Results: The mean difference of trueness was 26.61 ± 5.07 µm with the suggested strategy of intraoral scanning and using a new design of the tip, 37.99 ± 4.94 µm with the suggested strategy of intraoral scanning and using the old design of the tip, and 51.22 ± 6.57 µm with a new strategy of intraoral scanning and using the old design of the tip. The mean difference of precision was 23.57 ± 5.77 µm with the suggested strategy of intraoral scanning and using a new design of the tip, 38.34 ± 11.39 µm with the suggested strategy of intraoral scanning and using the old design of the tip, and 46.93 ± 7.15 µm with a new strategy of intraoral scanning and using the old design of the tip. No difference was found in the trueness and precision data extracted using the two different programs of superimposition Geomagic Control X and Medit Compare. Conclusions: The outcomes of this study showed that the latest version of I-Medit 500 with the use of a new tip seems to be promising in terms of accuracy when a full arch needs to be scanned. Moreover, Medit Compare, which is an application of Medit IOS software, can be used to calculate IOS accuracy.

## 1. Introduction

The use of digital devices is strongly influencing the dental rehabilitation workflow both for single-crown rehabilitation and for full-arch prosthetic treatments [1].

Accurate fitting of the final restoration plays a key role in prosthetic rehabilitations’ long-term success [2].

The achievement of a passive fit restoration is strictly dependent on the master cast’s accuracy of the impression [2,3,4].

There are two ways to take an impression to produce a master cast: a conventional impression procedure and a digital one [2,3,4].

The digital impression procedure using an intraoral scanner (IOS) was introduced a few years ago [1,2,3,4].

Using an IOS instead of the conventional impression procedure provides several advantages such as reduction in the patient’s discomfort (especially for those with an accentuated gag reflex), stone cast pouring and storing, easier patient communication, and, last but not least, faster and cheaper communication between the dental clinic and the lab technician, as the impressions are forwarded via email [5,6,7,8,9].

If, on the one hand, IOS accuracy overlaps with the conventional impression in short-span rehabilitation, on the other hand, its usage becomes questionable when large-span arches need to be treated [5,6,7,8,9].

Although there is not yet a scientific consensus, a value of 100 µm has been reported by the literature as an acceptable value in terms of prosthesis misfit [10,11,12].

The latest IOS systems seem to be promising in terms of accuracy (trueness and precision) even when full arches are scanned.

To assess the IOS’s accuracy, the software works by making a comparison between the IOS STL file and the reference scan’s STL file using the best-fit algorithm function; this matching generates linear deviations between the two datasets that can be measured [13,14,15].

A lot of variables can influence the accuracy of an IOS: ambient light, distance, substrate, preparation design, scanning strategy, experience, different non-contact optical technologies, hardware, and software version [16,17,18,19,20,21,22].

Even if the latest studies reported impressive results on accuracy (much less than 100 µm) of scanning full dental arches, it is important to underline that all these studies are based on in vitro models; this means that in clinical situations with the presence of saliva and the patient’s tongue, the physical movement of the patient, and the different refractive index of the filling or restorative materials and the mirroring, accuracy could be affected dramatically [22,23,24,25,26,27].

It becomes a must to understand how some variables can influence accuracy in order to minimize the risk of an inaccurate digital model.

Accuracy is defined (ISO 5725-1) in terms of trueness and precision.

Trueness is the deviation of the object scanned with an IOS from its real geometry (in this case the object scanned with an extraoral scanner); this means that accuracy is described by the mean of the linear discrepancies between the object scanned with an extraoral scanner (reference) and the IOS scans of the object (target).

Precision represents the deviation between the repeated scans of the same object performed with the same IOS and with the same parameters. In other words, it represents how much a measurement could be systematically repeated. Precision is described by the mean of the discrepancies among the various scans of the object carried out with the IOS.

The discrepancies between the two scans (reference and target) are calculated overlapping the STL files derived from the two scans in dedicated software and using the best-fit algorithm function.

The linear distances between the reference scan dataset and the IOS dataset could be positive or negative; this depends on whether the surface points are outside or inside the reference scan. In this way, a deviation analysis in this software works as a normal vector deviation principle. This means that it is possible to compare the length of the normal vectors (vectors between the corresponding meshes), and based on the length and the direction (positive or negative) of the vector, the map changes color. A report based on the map’s color usually exhibits the deviation values.

It becomes crucial to clarify, as reported by Oh et al., that the use of these linear distances needs to be taken without their signs (absolute values) in order to calculate trueness and to avoid results cancelling each other [16].

There are several methods to eliminate the sign of these distances; it is possible to use the RMS, the ABS AVG, and the (90°–10°)/2 method.

All these methods are used in the literature to calculate the accuracy of an IOS, and all the values are referred to trueness if they result from the overlapping between the IOS scans and the lab scans or to precision if they result from the overlapping between the IOS scans with each other.

Even if it seems clear that the studies regarding the accuracy of an IOS have to report which of the above-mentioned values and which software to calculate these values are used, not all the studies clarify these points.

The aim of the present study was to evaluate the accuracy of an IOS (Medit I-500) in the full dental arch, in terms of the (90°–10°)/2 method, by using two different tips and two different scan strategies.

The second aim was to evaluate whether using two different software programs of overlapping could affect the accuracy values.

The null hypothesis was that no statistical difference would be found between the different tips or strategies used and no statistical difference would be found between the software programs used.

## 2. Materials and Methods

Ten type IV (Elite Rock, Zermarck S.P.A., Badia Polesine (RO), Italy) stone casts (5 maxillary and 5 mandibular) were produced and used for this study.

All these models were full dental arches of real patients recently treated.

All models were scanned with lab scans (Medit T710, Seoul, South Korea), exported in standard tessellation file format to be used as a reference dataset, and saved as a Reference Group (RG) from RG1 to RG 10.

The Medit T710 lab scan works with a four 5.0 MP camera system and blue-light scanning technology, ensuring significant accuracy (<4 µm).

An IOS (Medit I-500, version 2.3.5, Seoul, South Korea) and two different tips (old tip and new tip) were also used to scan the models.

The scan parameters were set up as suggested by the manufacturer, with blue-light mode, a filtering level of 2, and a focal length of 17 mm.

Not more than 5 scans were conducted on the same day in order to avoid the potential effect of the operator’s fatigue.

All the IOS scans were performed by the same operator (LVS), who has 2 years of experience with the Medit IOS.

The calibration of the IOS was performed before each scan, as suggested by the manufacturer.

The suggested scan strategy followed for the maxillary/mandibular model started from the posterior left area by scanning the occlusal side, and then it proceeded toward the posterior right area by following a zigzag movement when the anterior teeth (from 2.3 to 1.3) were scanned. After the occlusal area, the palatal/lingual side was scanned from the right to the left area, and then the buccal side was scanned from the posterior left area to the posterior right one (Figure 1).

The second scanning strategy followed for the maxillary/mandibular model started from the left canine with a zigzag movement until the controlateral canine and was proceeded by scanning the occlusal side until the right posterior area. Then, the palatal/lingual side was scanned from the posterior right area to the left canine, and finally, the buccal side was scanned from the left canine to the right posterior side. The same procedures were repeated by starting from the right canine, proceeding to the left posterior side (Figure 2).

Two different tips (old tip and new tip) and the first strategy were used to scan the model with the IOS; the old tip was also used to scan the model with the second strategy. These three different methods of scanning were called old tip, new tip, and second strategy.

Each model was scanned 10 times with the IOS following each method and post-processed by using the Fill Major Holes tool, exported in standard tessellation file format to be used as a target dataset, and saved as TG 1, 10 new tip; TG 1, 10 old tip; and TG 1, 10 second strategy (target group). The resulting STL files (numbered from TG1 to TG10 for each model and for each method used) were compared with the corresponding STL files obtained with lab scanning. Therefore, 300 scans were collected with the IOS.

### 2.1. Trueness and Precision

The procedure adopted to evaluate the trueness and precision of the IOS is here reported according to ISO 5725, in which a combination of such parameters is adopted to describe the accuracy of the measurement method. Before proceeding with the comparisons, all the STL files were processed to cut and trim the 3D scans, filling the remaining holes, and eliminate any possible mesh artifacts. The 3D models were uploaded in one of the most popular dental CAD software programs, usually applied for prosthesis designing, with mesh boundaries roughly cut in order to leave a few millimeters of gum.

Specifically, the first imported STL file was the reference one (RG1), and then the target STL files were imported and aligned to that, one by one, TG1,1 to TG10,1 for each method (TG new tip, TG old tip, TG second strategy). Due to the boundaries’ difference among the meshes, for the comparison group, a cut of the flanges was made along a path parallel to the muco-gingival line, 2 to 5 mm far from the gingival margin.

The same operation was repeated for all the remaining reference 3D models in order to export a total of 10 STL files belonging to the RG, together with the 90 STL files belonging to the TG. It is important to note that a random displacement was applied to every 3D model before the final exportation in order to start the following comparison procedure by adopting not-aligned 3D models.

Subsequently, the STL files were imported into two 3D inspection software programs, (Geomagic Control X, Geomagic, Morrisville, NC, USA) and Medit Compare (Seoul, South Korea), to evaluate the distance gap between the reference models and the target ones. 

As shown in Figure 3, the adopted procedure consisted of three different steps followed by a post-processing analysis performed on MATLAB r2019a. The first step consisted of importing the reference 3D model in Geomagic Control X software and setting the reference 3D model as “reference data” and the target model as “measured data.” The superimposition process started with an initial rough alignment that was required to perform the final most accurate alignment, based on the best-fit algorithm. This algorithm is based on the standard iterative close point (ICP) method in which the mesh deviation between the measured point data and the reference point data is minimized. The iterative process is based on the least-squares method where the sums of squares of mesh deviations between the processed data are minimized over all the rigid motions that could realign the two meshes.

Once the superimposition was finalized, the distances between the reference and target models were evaluated by performing a 3D compare. The deviations between the meshes were represented with a colormap surface. The greater positive and negative distances were indicated with red and blue colors, respectively.

Since the adopted software was able to evaluate exclusively some of the statistical parameters investigated in this paper, a MATLAB user-code script was developed to process the distance gaps exported from Geomagic Control X in.csv format. The data distances were imported in MATLAB as raw data that were first sorted from the highest to the lowest distance and then processed to evaluate the following parameters: minimum distance, maximum distance, average (AVG), absolute average (ABS AVG), root mean square (RMS), standard deviation (SD), variance, positive average (ABS+), negative average (ABS−), and (90°–10°)/2, where 90° is the 90th percentile and 10° is the 10th percentile.

The values resulting from the superimposition of the intraoral scan datasets (TG) on the reference scan datasets (RG), for a total of 300 measurements for each parameter calculated, were referred to as trueness values.

The values resulting from the superimposition of the intraoral scan datasets (TG) on each other were referred to as precision values. In this case, the first dataset of the RG was selected to calculate precision, so that 90 measurements were used for each parameter.

### 2.2. Medit compare Software

The overlapping procedure was repeated by using the software Medit Compare that is included in I-500 software version.

The RG data STL files and all the TG STL files belonging to the same RG and to the same scan procedure used were uploaded into the software (Figure 4); by using a specific tool each STL file of the TG was aligned to that of the RG (Figure 5), and the software automatically reported the following values for trueness (obtained from the overlapping of the scan dataset of the RG with the scan dataset of the TG) and precision (calculated by the intra-group overlapping): minimum distance, maximum distance, average (AVG), absolute average (ABS AVG), root mean square (RMS), standard deviation (SD), variance, positive average (ABS+), negative average (ABS−), and (90°–10°)/2, where 90° is the 90th percentile and 10° is the 10th percentile (Figure 6).

Only the values of the (90°–10°)/2 method were taken into account for the accuracy evaluation of the IOS.

### 2.3. Statistical Analysis

A database was created using Microsoft Excel (Microsoft, Redmond, WA, USA). Descriptive statistics including mean ± SD values were calculated for each variable, and box plots were used to evaluate data outliers. The Shapiro–Wilk test was used to determine whether the data conformed to a normal distribution.

One-way ANOVA was used to assess the difference in trueness and precision between two different strategies of intraoral digital impression and a new scanning tip.

Pairwise comparisons were performed with Tukey’s HSD correction for multiple comparisons.

The independent-samples t-test was carried out to identify statistically significant differences in trueness and precision obtained from two different programs of superimposition: Geomagic Control X and Medit Compare. 

Data were evaluated using the standard statistical analysis software Statistical Package for the Social Sciences (SPSS), version 20.0 (IBM Corporation, Armonk, NY, USA). In each test, the cutoff for statistical significance was *p* ≤ 0.05.

## 3. Results

A total of 300 scans (100 with the old strategy of digital impression, 100 with the new one, and 100 with the old strategy using a new scanning tip) were used to evaluate the difference in trueness and precision comparing two different types of digital impression techniques and a new scanning tip for easier intraoral access and whether the results could be influenced by the programs used for the superimposition.

No outliers were detected, and the assumption of normality, assessed by the Shapiro–Wilk test, was not violated.

The mean difference of trueness was 26.61 ± 5.07 µm with the suggested strategy of intraoral scanning and using a new design of the tip, 37.99 ± 4.94 µm with the suggested strategy of intraoral scanning and using the old design of the tip, and 51.22 ± 6.57 µm with a new strategy of intraoral scanning and using the old design of the tip (F = 48.829; *p* < 0.001) (Figure 7). Post hoc analysis with Tukey’s HSD correction revealed that the mean difference of trueness was statistically significantly lower with the suggested strategy and using a new tip compared to the suggested strategy of scanning with the old design of the tip (*p* < 0.001) and a new strategy of scanning using the old design of the tip (*p* < 0.001) (Table 1).

The mean difference of precision was 23.57 ± 5.77 µm with the suggested strategy of intraoral scanning and using a new design of the tip, 38.34 ± 11.39 µm with the suggested strategy of intraoral scanning and using the old design of the tip, and 46.93 ± 7.15 µm with a new strategy of intraoral scanning and using the old design of the tip (F = 86.032; *p* < 0.001) (Figure 7). Post hoc analysis with Tukey’s HSD correction revealed that the mean difference of precision was statistically significantly lower with the suggested strategy and using a new tip compared to the suggested strategy of scanning with the old design of the tip (*p* < 0.001) and a new strategy of scanning using the old design of the tip (*p* < 0.001) (Table 1).

No difference was found in the trueness (F = 5.830; *p* = 0.259) and precision (F = 0.652; *p* = 0.113) data extracted using the two different programs of superimposition, Geomagic Control X and Medit Compare (Figure 8) (Table 2).

## 4. Discussion

The first aim of this in vitro study was to evaluate the accuracy of an IOS (Medit I-500) in the full dental arch by using two different tips and two different scan strategies. The null hypothesis that no statistically difference would be found between the different tips or strategies used was rejected. In this study, trueness was analyzed by overlapping the scan dataset made with Medit I-500 with the scan dataset obtained with lab scanning, and the values of the (90°–10°)/2 method were reported (the highest and lowest 10% of the surface was not taken into account because of the margin’s effect and the different scan sizes of the models).

Precision was evaluated by using the same values of trueness coming from the intra-group overlapping (scan dataset made with the IOS scans overlapped and compared with each other).

The use of a new tip considered in this study showed high values of accuracy in terms of trueness and precision; moreover, the suggested strategy scan resulted in better accuracy results than a second strategy.

Other studies have evaluated the accuracy of Medit I-500 in full dental arches. Ender et al. [18] made 10 scans of a complete dentate arch by using one of the first versions of Medit I-500 with the recommendation of the manufacturer and using GOM Inspect (2018rev. 114010) as the 3D software for comparison. They used the (90°–10°)/2 percentile method values for the calculation of trueness (comparisons with reference scans; *N* = 10) and precision (intra-group comparisons; *N* = 45). They reported an accuracy of 88.9 µm for trueness and 56.1 µm for precision, and they did not specify whether they left out the gum.

The trueness and precision in the present study were much better than Ender et al.’s [18], and a possible reason could be that their study was conducted by using one of the first versions of Medit I-500 software.

Lee et al. [28] used Carestream 3600 and Medit I-500 in order to evaluate the precision of IOSs by scanning several times five different models (one represented a full-arch dentate model, the other four partially edentulous models with the missing teeth differing in position and in number among the models). They used Geomagic Control X as software for the 3D comparison and the RMS as a value to measure precision. Passing from the full-arch dentate model to the partially edentulous ones (with the smoother surface scanned), the RMS showed a higher mean and a larger range of interval values (the mean and the interval range showed an increment in the models with more missing teeth). The precision value of the RMS for the full dental arch was 52.30 µm (SD 14.99 µm). These results could not be compared with the ones of this paper, because Lee et al. used the RMS and not the (90°–10°)/2 percentile method to calculate the precision.

Michelinakis et al. [29] scanned 38 models with Medit I-500 (no indications of which version was used) according to the manufacturer’s recommendations. The software used for 3D overlapping was CloudCompare (version 2.11 alpha; Anoia). Trueness was calculated by using absolute average values (15.8 µm). For the calculation of the precision, one model was chosen and scanned 10 times. The mean of the standard deviation between the intra-group analysis was used for precision evaluation (20 µm).

In our study, we used the (90°–10°)/2 method for trueness and precision so that our values cannot be compared with the ones of Michelanikis et al.(Table 3).

Regarding the scanning strategy, Ender et al. [28] found that the scan strategy does not affect the accuracy in short-span segments, such as a sextant, but it exerts an influence when a large span is considered (a full dental arch).

The results of Ender et al. are consistent with those of Mennito et al. [29]; they evaluated the influence of five scan strategies on accuracy and reported that the scan pattern influences trueness and precision when larger areas are scanned.

Latham et al. [30] used four IOSs and four different scan strategies in order to evaluate the effect of the scan pattern on trueness and precision when a full dental arch model was scanned. The scan pattern exerted an influence on trueness in only one of the four IOSs and in two IOSs when the precision values were considered.

Medina-Sotomayor [31] analyzed four scanning strategies by using four IOSs (Trios, Itero, Omnicam, and True Definition) in order to evaluate whether one of them exerts a difference in terms of accuracy using a large-span model as reference. They reported no statistically difference for three IOSs (Trios, Omnicam, and True Definition), but Itero showed better results with one of the strategies used. Discrepancies were analyzed using Geomagic Control X (Geomagic, Morrisville, North Carolina, USA, 2013). The range of trueness was 52.95 to 57.09 for Trios, 78.77 to 109.29 for iTero, 94.08 to 108.52 for Omnicam, and 28.78 to 35.57 for True Definition.

Müller et al. [32] analyzed three different strategies (A, B, C) using the Trios Pod scanner, and they reported that strategy B (first occlusal-palatal, return buccal) is more accurate in terms of trueness (but no statistically significant difference was reported) and in terms of precision (a statistically significant difference was reported). The trueness (mean ± standard deviation) was 17.9 ± 16.4 μm for strategy A, 17.1 ± 13.7 μm for strategy B, and 26.8 ± 14.7 μm for strategy C without a statistically significant difference. Precision was lowest for strategy A (35.0 ± 51.1 μm) and significantly different than strategy B (7.9 ± 5.6 μm) and strategy C (8.5 ± 6.3 μm). The values within the 90/10 percentiles from the digital superimposition were used.

Oh et al. [16] investigated the effects of rotation of the IOS on the accuracy when a full dental arch was scanned.

They used two IOSs, TRIOS 3 and Medit I-500, and three different scanning strategies (two continuous scan strategies, with one of them maintaining the IOS horizontally (CH strategy) and one by rotating (CV strategy) the IOS in a vertical direction when the anterior teeth were scanned; the last strategy was segmental (S) with the anterior region of the teeth scanned first, followed by the left and right posterior areas). The values of trueness were statistically worse only for the CV strategy. No differences were found in precision for the different scanning strategies.

In our study, as the ones reported above, we found that the use of a second strategy affects the results of accuracy so that it appears clear that it becomes a must to specify which strategy is used when different studies of an IOS accuracy are compared.

The improvements in the hardware of the new tip justify, probably, the better results when using it instead of the old one; regarding the comparison between the suggested strategy and the second one, the better results of the first one are, probably, because the scan of the surfaces with a continuous movement without any interruption of the surface scan introduces fewer distortions during software acquisition.

The second aim of this study was to evaluate whether the use of two different software programs (Geomagic Control X and Medit Compare) to calculate accuracy could affect the results; both software programs produced consistent results.

The limitations of this study were that this was an in vitro study and the IOS scans were made by only one operator. Further studies are required to standardize the method that should be used to evaluate IOS accuracy.

Moreover, in this in vitro study, Geomagic 3D software and Medit Compare were used; an additional study should be carried out in order to evaluate whether there would be some differences in terms of the reported values if a different 3D comparison software program is used, as well the use of Cloud Compare and GOM Inspect.

## 5. Conclusions

Even if there is a convergence in the methods to be used for accuracy evaluation, there is not a general consensus on which values or software need to be used to calculate them. In addition, different accuracy values were used and compared with each other, hence potentially generating bias. It should be mandatory to clearly specify the methods and the values considered in order to enable studies’ comparison and a better understanding for the readers.

The outcomes of this study showed that the latest version of Medit I-500 with the use of a new tip seems to be promising in terms of accuracy when a full dental arch needs to be scanned.

Moreover, Medit Compare, which is an application of Medit IOS software, can be used to calculate IOS accuracy.

## Figures and Tables

**Figure 1 ijerph-18-09946-f001:**
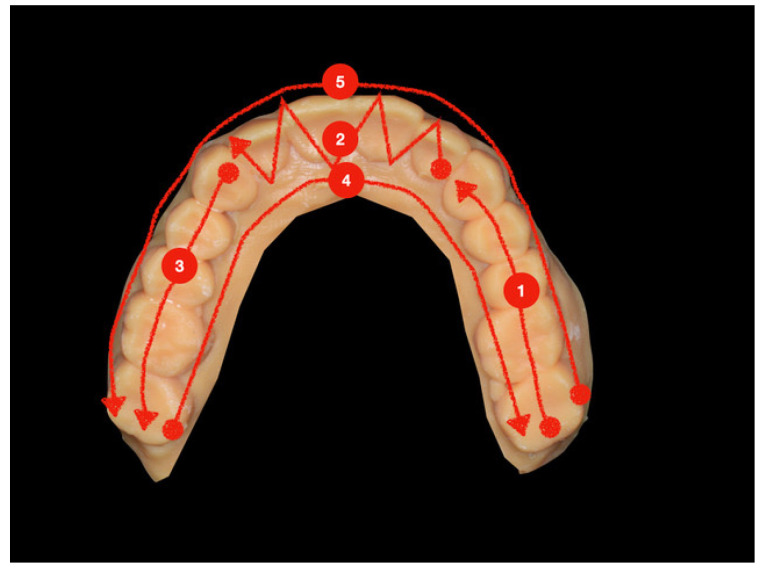
The path of the scan in the suggested strategy.

**Figure 2 ijerph-18-09946-f002:**
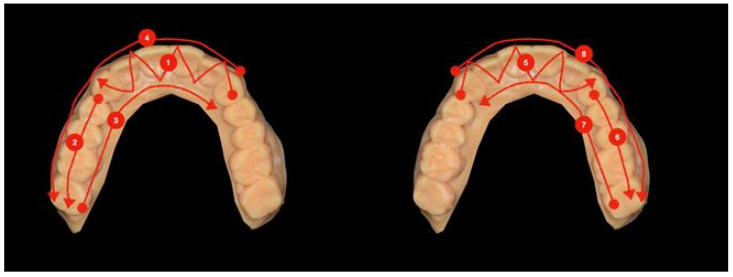
The path of the scan in the second strategy.

**Figure 3 ijerph-18-09946-f003:**
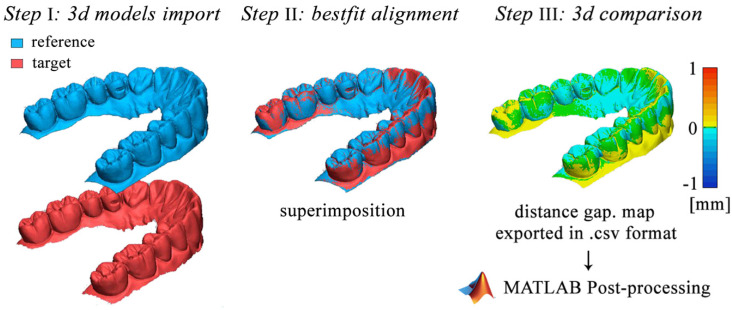
Example of a 3D comparison procedure performed by using Geomagic’s software.

**Figure 4 ijerph-18-09946-f004:**
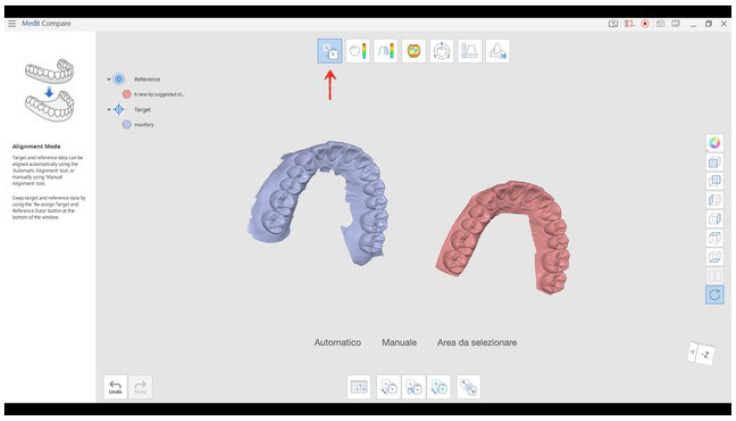
The STL file of the RG and that of the TG were uploaded into the software. The arrow indicates the tool to be used for the alignment of the two STL files.

**Figure 5 ijerph-18-09946-f005:**
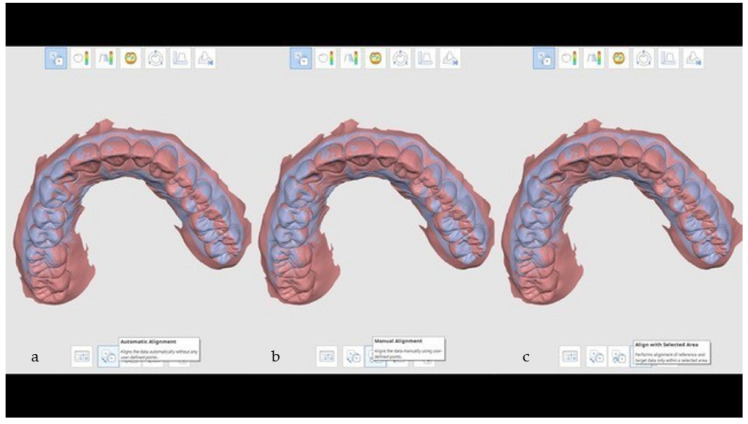
Three different methods can be used to align STL files: (**a**) automatic, (**b**) manually by clicking similar points on the two STL files, and (**c**) alignment with selected areas, in which only the selected areas were aligned.

**Figure 6 ijerph-18-09946-f006:**
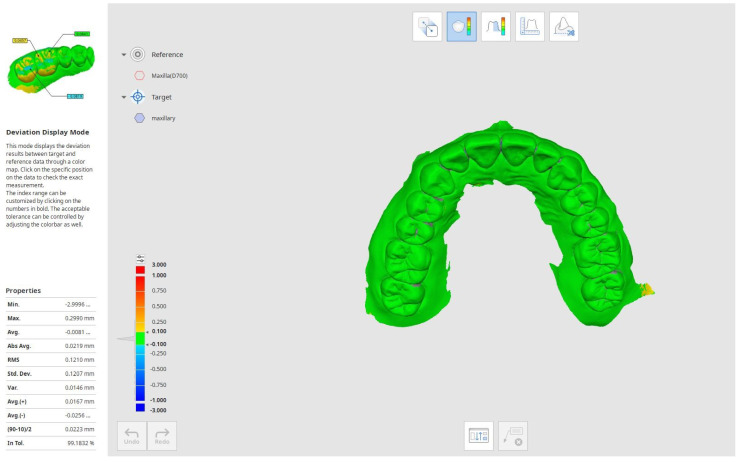
Medit Compare automatically calculates and shows in a column the deviations between the reference scan made with a lab scan and the target one made with an IOS scan.

**Figure 7 ijerph-18-09946-f007:**
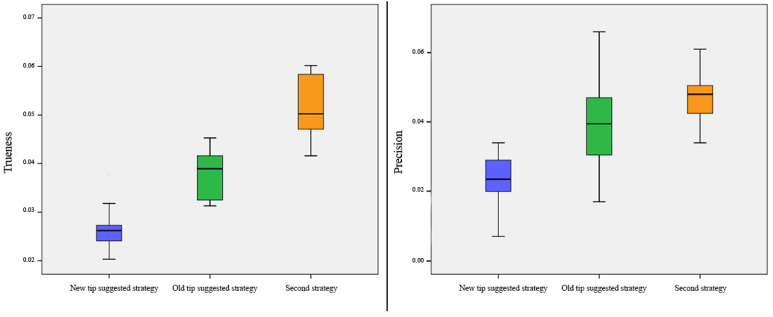
Box plots showing the median, quartile, and minimum and maximum values of the mean differences of trueness and precision between two different intraoral scanning procedures and two different tips used.

**Figure 8 ijerph-18-09946-f008:**
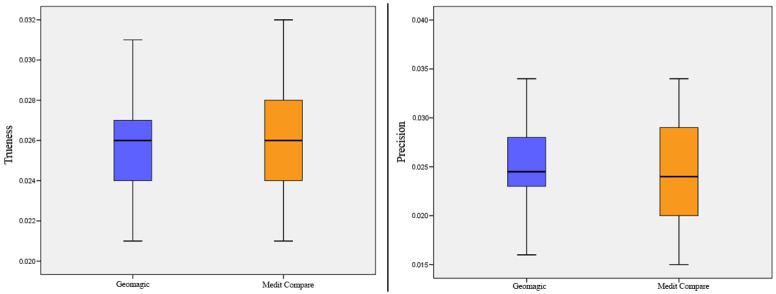
Box plots showing the median, quartile, and minimum and maximum values of the mean differences of trueness and precision calculated between two different software programs of superimposition.

**Table 1 ijerph-18-09946-t001:** Mean difference in trueness and precision between two different strategies of intraoral digital impression and a new scanning tip.

**Trueness (µm)**						
**(I) Model**	**(J) Model**	**Mean Difference (I–J)**	**Std. Error**	***p*-Value**	**95% Confidence Interval**	
					**Lower Bound**	**Upper Bound**
New tip, suggested strategy	Old tip, suggested strategy	−11.38 *	2.49	0.000	−17.56	−5.19
	Second strategy	−24.61 *	2.49	0.000	−30.79	−18.42
Old tip, suggested strategy	New tip, suggested strategy	11.38 *	2.49	0.000	5.19	17.56
	Second strategy	−13.23 *	2.49	0.000	−19.41	−7.04
Second strategy	New tip, suggested strategy	24.61 *	2.49	0.000	18.42	30.79
	Old tip, suggested strategy	13.23 *	2.49	0.000	7.049	19.41
**Precision (µm)**						
**(I) Model**	**(J) Model**	**Mean Difference (I–J)**	**Std. Error**	***p*-Value**	**95% Confidence Interval**	
					**Lower Bound**	**Upper Bound**
New tip, suggested strategy	Old tip, suggested strategy	−14.77 *	1.80	0.000	−19.04	−10.50
	Second strategy	−23.36 *	1.80	0.000	−27.63	−19.09
Old tip, suggested strategy	New tip, suggested strategy	14.77 *	1.80	0.000	10.50	19.04
	Second strategy	−8.59 *	1.80	0.000	−12.86	−4.31
Second strategy	New tip, suggested strategy	23.36 *	1.80	0.000	19.09	27.63
	Old tip, suggested strategy	8.59 *	1.80	0.000	4.31	12.86

* The mean difference is significant at the 0.05 level.

**Table 2 ijerph-18-09946-t002:** Mean difference in trueness and precision between two different programs of superimposition (Geomagic Control X and Medit Compare).

**Trueness (µm)**						
**t**	**df**	***p*-Value**	**Mean Difference**	**Std. Error Difference**	**95% Confidence Interval of the Difference**	
					**Lower**	**Upper**
−1.132	198	0.259	−0.0003900	0.0003445	−0.0010694	0.0002894
**Precision (µm)**						
**t**	**df**	***p*-Value**	**Mean Difference**	**Std. Error Difference**	**95% Confidence Interval of the Difference**	
					**Lower**	**Upper**
1.602	80	0.113	0.0015778	0.0009846	−0.0003816	0.0035371

**Table 3 ijerph-18-09946-t003:** Different accuracy values, in terms of trueness and precision, reported by several authors, when a full dental arch is scanned by Medit I-500.

Author	Software Used	Trueness	Precision
Ender A.	GOM Inspect	(90°–10°)/2 Percentile = 88.9 µm	(90°–10°)/2 Percentile = 56.1 µm
Lee J.H.	Geomagic Control X	Not reported	RMS = 52.30 µm
Michelanikis G.	CloudCompare	Absolute average = 15.8 µm	Absolute average = 20 µm
Stefanelli L.V.	Geomagic Control X	(90°–10°)/2 Percentile = 26.61 µm	(90°–10°)/2 Percentile = 23.57 µm
Stefanelli L.V.	Medit Compare	(90°–10°)/2 Percentile = 26.24 µm	(90°–10°)/2 Percentile = 24.26 µm

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
