# Peer review of "Use of Intraoral Scanners for Full Dental Arches: Could Different Strategies or Overlapping Software Affect Accuracy?"

_ijerph, 2021, doi:10.3390/ijerph18199946_

Round 1

Reviewer 1 Report

In this article, authors used different tips and scan strategies for intra oral scanners to compared the trueness and precisions. Authors also compared the results calculated from two overlapping software (Medit Compare and Geomagic Control X). These works are well designed and have a good clinical significance. However, there are some flaws should be improved:

  1. Sketch maps should be added to show the two scanning strategies.
  2. In this research, authors configured three combinations of parameters: old tip and first scanning strategy, new tip and first scanning strategy, old tip and second scanning strategy. Why new tip and second scanning strategy omitted?
  3. In section 2.1, authors should detail the method of calculation of trueness and precisions
  4. In abstract and results sections, authors mentioned suggested strategy and second strategy, but in materials and methods section, author introduced first strategy and second strategy, so the first strategy is the suggested strategy? Please make clarification in the materials and methods section.
  5. The manual alignment and selected area alignment are not necessary in Fig.3 and may cause misunderstanding, since these methods were not used in this research.
  6. In the results section, tables are preferred to show the trueness and precision values for each group, and the results of statistic analysis should also be presented, including F values, P values, etc.
  7. In Fig.5 groups with significant different values should bettor be noted.
  8. In discussion section, authors should try to explain why suggested strategy and new tip achieved better results.
  9. There are some mistypings:
    page 2, line 63: then -> than
    page 3, line 123: then -> than

Author Response

Reviewer 1

Comments and Suggestions for Authors

In this article, authors used different tips and scan strategies for intra oral scanners to compared the trueness and precisions. Authors also compared the results calculated from two overlapping software (Medit Compare and Geomagic Control X). These works are well designed and have a good clinical significance. However, there are some flawsshould be improved:

  1. Sketch maps should be added to show the two scanning strategies.

Answer: A new figure has been added.

  1. In this research, authors configured three combinations of parameters: old tip and first scanning strategy, new tip and first scanning strategy, old tip and second scanning strategy. Why new tip and second scanning strategy omitted?

Answer:the new tip is an experimental device that was used to be tested for the first time in this study, so that we decided to use it only with suggested strategy in order to reduce the variables that could affect the results. Its usage with several scanning strategies could be object of a separate study

  1. In section 2.1, authors should detail the method of calculation of trueness and precisions

Answer:The details for the trueness and precision calculation have been added in the section 2.1

  1. In abstract and results sections, authors mentioned suggested strategy and second strategy, but in materials and methods section, author introduced first strategy and second strategy, so the first strategy is the suggested strategy? Please make clarification in the materials and methods section.

Answer: The first strategy is the suggested strategy. The terms suggested strategy were inserted in materials and methods instead of the first strategy.

  1. The manual alignment and selected area alignment are not necessary in Fig.3 and may cause misunderstanding, since these methods were not used in this research.

Answer:The manual alignment and selected area alignment have been cancelled from the figure.3

  1. In the results section, tables are preferred to show the trueness and precision values for each group, and the results of statistic analysis should also be presented, including F values, P values, etc.

Answer:As suggested, the manuscript has been changed

  1. In Fig.5 groups with significant different values should bettor be noted.

Answer:The significant different values have been reported in table 1 and table 2

  1. In discussion section, authors should try to explain why suggested strategy and new tip achieved better results.

Answer:A possible explanation has been added in the discussion

  1. There are some mistypings:
    page 2, line 63: then -> than
    page 3, line 123: then -> than

Answer: The correction has been done

Reviewer 2 Report

-        Introduction: it should be reminded how performances are different between laser scanners based on triangulation (Optics and Lasers in Engineering 54 (2014) 203–221) or on Time of Flight (Optics and Lasers in Engineering 54 (2014) 187–196)

-        Methods: Please report more details about alignment algorithms implemented by cited software (is any of them based on Procrustes analysis (Applied Sciences , 2021, 11(11), 5204)?

-        Figure 5-6: labels are too small

-        P9 R304-306: having supposed a normal distribution 90%  should correspond to the average + 1.28 sd

- P9R332: there is a typo: ‘range’

- P9R336-340: A/B/C scan strategie would be declared at the beginning of paragraph (row 336: three different strategies A/B/C...

- P10R356: i think the word "MUST" would be instead of "MAST"

- End of discussion: a figure comparing accuracy declared by various authors would be very helpful

Author Response

Reviewer 2

Comments and Suggestions for Authors

-        Introduction: it should be reminded how performances are different between laser scanners based on triangulation (Optics and Lasers in Engineering 54 (2014) 203–221) or on Time of Flight (Optics and Lasers in Engineering 54 (2014) 187–196)

Answer: the influence of different non-contact optical technologies as parameter on accuracy has been reminded in the introduction and the two suggested papers added into the references

-        Methods: Please report more details about alignment algorithms implemented by cited software (is any of them based on Procrustes analysis (Applied Sciences , 2021, 11(11), 5204)?

Answer:This algorithm is based on the standard iterative closes point (ICP) method in which the mesh deviation between the measured point data and the reference point data is minimized. The iterative process is based on the least-squares method where the sums of squares of mesh deviations between the processed data are minimized over all the rigid motions that could realign the two meshes.

-        Figure 5-6: labels are too small

As suggested, the figures have been modified

-        P9 R304-306: having supposed a normal distribution 90%  should correspond to the average + 1.28 sd

The Tests of normal distribution for all analysis were reported in the tables below.

Tests of Normality

Programs_trueness

Kolmogorov-Smirnova

Shapiro-Wilk

Statistic

df

Sig.

Statistic

df

Sig.

Trueness_with_different_programs

Geomagic

.102

100

.012

.977

100

.075

Medit Compare

.094

100

.028

.975

100

.052

a. Lilliefors Significance Correction

Tests of Normality

Programs_precision

Kolmogorov-Smirnova

Shapiro-Wilk

Statistic

df

Sig.

Statistic

df

Sig.

Precision_with_different_programs

Geomagic

.166

44

.004

.974

44

.403

Medit Compare

.111

38

.200*

.967

38

.315

*. This is a lower bound of the true significance.

a. Lilliefors Significance Correction

Tests of Normality

Model

Kolmogorov-Smirnova

Shapiro-Wilk

Statistic

df

Sig.

Statistic

df

Sig.

Accuracy_of_intraoral_digital_scans

New tip suggested strategy

,246

10

,088

,903

10

,237

Old tip suggested strategy

,167

10

,200*

,926

10

,407

Second strategy

,163

10

,200*

,934

10

,493

*. This is a lower bound of the true significance.

a. Lilliefors Significance Correction

Tests of Normality

Tip_and_strategy

Kolmogorov-Smirnova

Shapiro-Wilk

Statistic

df

Sig.

Statistic

df

Sig.

Precision

New tip suggested strategy

.122

44

.097

.966

44

.224

Old tip suggested strategy

.077

44

.200*

.981

44

.666

Old tip second strategy

.084

44

.200*

.968

44

.253

*. This is a lower bound of the true significance.

a. Lilliefors Significance Correction

- P9R332: there is a typo: ‘range’

Answer:The correction has been done

- P9R336-340: A/B/C scan strategie would be declared at the beginning of paragraph (row 336: three different strategies A/B/C...

Answer:The The correction has been done as suggested

- P10R356: i think the word "MUST" would be instead of "MAST"

Answer:The correction has been done

- End of discussion: a figure comparing accuracy declared by various authors would be very helpful

Answer: a table comparing the accuracy reported by different authors using medit I-500 in the full dentate arches was added